# Targeted Therapies and Immune-Checkpoint Inhibition in Head and Neck Squamous Cell Carcinoma: Where Do We Stand Today and Where to Go?

**DOI:** 10.3390/cancers11040472

**Published:** 2019-04-03

**Authors:** Jens von der Grün, Franz Rödel, Christian Brandts, Emmanouil Fokas, Matthias Guckenberger, Claus Rödel, Panagiotis Balermpas

**Affiliations:** 1Department of Radiation Oncology, Theodor-Stern-Kai 7, University of Frankfurt, 60590 Frankfurt, Germany; Jens.vonderGruen@kgu.de (J.v.d.G.); Franz.Roedel@kgu.de (F.R.); Emmanouil.Fokas@kgu.de (E.F.); ClausMichael.Roedel@kgu.de (C.R.); 2Frankfurt Cancer Institute (FCI), Theodor-Stern-Kai 7, University of Frankfurt, 60590 Frankfurt, Germany; 3German Cancer Research Center (DKFZ), Im Neuenheimer Feld 280, 69120 Heidelberg, Germany; Christian.Brandts@kgu.de; 4German Cancer Consortium (DKTK), Partner Site: Frankfurt a. M., Theodor-Stern-Kai 7, University of Frankfurt, 60590 Frankfurt, Germany; 5Department of Medicine, Hematology/Oncology, University Cancer Center Frankfurt (UCT), Theodor-Stern-Kai 7, University of Frankfurt, 60590 Frankfurt, Germany; 6Department of Radiation Oncology, Rämistrasse 100, University Hospital Zurich, 8091 Zürich, Switzerland; Matthias.Guckenberger@usz.ch

**Keywords:** immune-checkpoint inhibition, targeted therapy, head and neck cancer, EGFR, mTOR, TKI

## Abstract

With an increased understanding of the tumor biology of squamous cell carcinoma of the head and neck (SCCHN), targeted therapies have found their way into the clinical treatment routines against this entity. Nevertheless, to date platinum-based cytostatic agents remain the first line choice and targeting the epidermal growth factor-receptor (EGFR) with combined cetuximab and radiation therapy remains the only targeted therapy approved in the curative setting. Investigation of immune checkpoint inhibitors (ICI), such as antibodies targeting programmed cell death protein 1 (PD-1) and its ligand PD-L1, resulted in a change of paradigms in oncology and in the first approval of new drugs for treating SCCHN. Nivolumab and pembrolizumab, two anti-PD-1 antibodies, were the first agents shown to improve overall survival for patients with metastatic/recurrent tumors in recent years. Currently, several clinical trials investigate the role of ICI in different therapeutic settings. A robust set of biomarkers will be an inevitable tool for future individualized treatment approaches including radiation dose de-escalation and escalation strategies. This review aims to summarize achieved goals, the current status and future perspectives regarding targeted therapies and ICI in the management of SCCHN.

## 1. Introduction

Squamous cell carcinoma of the head and neck (SCCHN) is diagnosed in approximately 500,000 patients per year worldwide with rising incidence [1], mainly attributed to younger individuals with Human-Papilloma-Virus (HPV)-positive tumors and to an increasing number of elderly patients due to an improved life expectancy [2]. Standard treatment for early stage tumors consists of surgery or radiotherapy (RT), and for locoregionally advanced tumors of radical surgery with subsequent adjuvant RT/chemoradiotherapy (CRT) or primary CRT [3]. Despite advances in CRT, locoregional recurrences occur in up to 40% and distant metastases in approximately 25% of all cases with locally advanced (LA) SCCHN [4,5,6]. To date, platinum-based chemotherapy remains first line systemic therapy both in the curative and recurrent and/or metastatic (R/M) setting. The approval of the epidermal growth factor-receptor (EGFR) antagonist cetuximab introduced the first targeted therapy in SCCHN showing increased locoregional control when compared to RT alone [7] and improved outcome when combined with chemotherapy in the palliative situation [8].

Interestingly, the immune system appears to affect treatment response in the context of RT/CRT [9]. Accumulating data regarding the interplay between tumor/microenvironment and the immune contexture, as well as mechanisms underlying immune-checkpoint pathway regulation, suggest that targeted therapies can promote anti-tumor immunity and mediate durable cancer regression [9]. RT can modulate these effects via “immune vaccination” (e.g., through up-regulation of MHC class I molecules) and enhanced antigen presentation on the surface of tumor and dendritic cells (DC), which may also partly explain the abscopal effect occasionally seen in the clinical setting [10]. These observations, together with the encouraging results observed for other malignant diseases [11,12] led to an abundance of clinical trials aiming to modulate immune response for SCCHN, especially regarding immune-checkpoint inhibition (ICI) [13,14,15].

This review discusses planned, recruiting and completed clinical trials assessing immuno- and targeted therapies in the primary and R/M setting for SCCHN. Included studies are prospective clinical landmark trials, which provided evidence for current treatment strategies, or major ongoing trials (cited according to their ClinicalTrials.gov registry number NCT). Trials were identified at www.ClinicalTrials.gov, at PubMed database or from presentations at corresponding societal meetings.

## 2. Immune-Checkpoint Inhibition in SCCHN: Mode of Action

### 2.1. PD-1/PD-L1 Axis and CTLA4 Blockade

The blockade of the PD-1/PD-L1 interaction and the cytotoxic T-lymphocyte antigen 4 (CTLA-4) via various antibodies has emerged as a powerful tool in anticancer therapy with the potential to reverse cancer-mediated immunosuppression. Among the different T-cell inhibitory and co-inhibitory pathways, those two have been best-characterized within recent years [16].

PD-1 was first described in 1992 and its role as an immune checkpoint has been unraveled in the following years [16,17]. Aside from T-cells, PD-1 is also expressed by natural killer (NK) cells, B-cells regulatory T-cells and macrophages. PD-1 is a transmembrane receptor inhibiting the T-cell receptor (TCR) downstream pathway via tyrosine phosphatase SHP-2 activity. Regarding T-cells it affects activation, exhaustion, immune tolerance and inflammation [18]. Activation of PD-1 occurs via binding of its two ligands PD-L1 (expressed by antigen-presenting cells (APC) and somatic cells) and PD-L2 (expressed by macrophages and DC). Both ligands can be overexpressed in SCCHN and are naturally induced by pro-inflammatory signals, protecting self-tissue from an excessive immune response [16,19] (Figure 1). Several antibodies against PD-1 (pembrolizumab, nivolumab) and PD-L1 (atezolizumab, durvalumab, avelumab) have already gained approval by the Food and Drug Administration (FDA) for various tumor entities.

CTLA-4 is a potent inhibitor of T-cell proliferation and activation [20,21,22], and its blockade via ipilimumab has been tested in clinical trials since 2000 to treat patients with melanoma and ovarian carcinoma [23,24]. Upon co-activation of the TCR and co-stimulatory CD28 receptor, CTLA-4 is expressed and translocates to the cell membrane. CTLA-4 competes with CD28 for its ligands CD80 and CD86 with a 50–2000 fold increased affinity [20]. Also, CTLA-4 can bind phosphatases to its intracellular part to further reduce TCR and CD28 signaling affecting both CD8+ and CD4+ T-cells [20,25]. Unlike PD-1, CTLA-4 is a non-redundant immune checkpoint and knock-out mice die within weeks due to T-cell mediated multi-organ inflammation [26,27] (Figure 2).

Radiotherapy modulates the host immune system in various ways such as an upregulation of MHC (major histocompatibility complex) class I molecules enhancing local CD8+ T-cell effects, intensified expression of calreticulin and other prophagocytic signals and upregulation of PD-L1 on the tumor cells surface [10]. This local tumor immune-vaccination by RT also promotes the systemic abscopal effect occasionally seen in the clinical setting [28]. The host immune response against the tumor prompted by RT could be enforced by either generally boosted immune response (anti-CLTA-4) or reversed immunoediting (e.g., anti-PD-1/PD-L1) [29,30]. Also, cytotoxic chemotherapy has important immunomodulatory activity and its effects cannot be reduced to immunosuppression via myelotoxicity. For example, platinum-based drugs enhance T-cell activation by dendritic cells and docetaxel decreases numbers of regulatory T-cells [31]. In line with that, the phase III randomized, Keynote-048 trial [15] has evaluated this concept (pembrolizumab + chemotherapy) and proven its enhanced efficacy in the clinical setting for head and neck cancer (see also Section 2.2.1).

A robust set of biomarkers will be of great importance to select patients for targeted therapies and combined approaches. Emerging predictive biomarkers for PD-1/PD-L1 blockade are micro-satellite instability (MSI-H), tumor mutational burden (TMB) and T cell-inflamed gene expression profile (GEP) [32]. So far, data on safety and efficacy of combined RT/CRT and ICI are mostly of retrospective nature [33].

### 2.2. Immune-Checkpoint Inhibition in the Recurrent/Metastatic Situation

#### 2.2.1. PD-1 Inhibition

For R/M SCCHN several immunoregulatory receptors and combined treatments are currently under investigation [34,35]. Pembrolizumab, an IgG4 PD-1 monoclonal antibody (MK-3475, Merck Sharp & Dohme), was introduced in 2013 within the Keynote-012 trial recruiting patients with different solid malignancies. Sixty patients with R/M SCCHN (+/− prior treatment) with any detectable PD-L1 expression (≥1%) on tumor cells/stroma and Eastern Cooperative Oncology Group (ECOG) performance status (PS) 0–1 were treated with pembrolizumab monotherapy and received 10 mg/kg bodyweight (BW) every two weeks (q2w). In total, 17% of the patients experienced grade 3–4 drug-related adverse events (AE, AEs covering pneumonitis, colitis, hepatitis, adrenal insufficiency, diabetes mellitus, and skin toxicities). Overall response rate (ORR) was 18% after 14 months of median follow-up [36]. The expansion cohort of the Keynote-012 trial comprised further 132 enrolled patients, regardless of their PD-L1 expression status treated with an adjusted dosage of pembrolizumab (200 mg fixed-dose (FD) q3w). Treatment-related grade 3–4 AEs occurred in 9% and the ORR after a median follow-up of 9 months was 18% and equal to that of the first phase. [37]. Pooled analysis revealed treatment-related AEs of any grade and grade 3–4 in 64% and 13% of the patients, respectively, but no treatment-related deaths occurred [38].

Amongst other early studies, the ECHO-202/Keynote-037 phase I/II trial enrolled patients with R/M disease with progress to prior therapy with different tumor entities, but only 2 patients with R/M SCCHN. Phase I results were published recently indicating that epacadostat + pembrolizumab were well tolerated and showed antitumor activity in different advanced solid tumors [39]. By contrast, the Keynote-669/ECHO-304 phase III trial testing the combination (pembrolizumab with the indoleamine 2,3-dioxygenase 1 enzyme inhibitor vs. EXTREME regimen) was closed after recruiting only 89 of originally 625 planned patients and no report has been published yet [40].

In the Keynote-055 phase II trial, 171 patients with disease progression within 6 months of platinum and cetuximab therapy were treated by pembrolizumab: 75% of the patients previously already had ≥2 lines of therapy for R/M disease. Combined positive score (number of PD-L1 stained cells including tumor cells, lymphocytes and macrophages/total number of viable tumor cells) ×100, CPS) for PD-L1 of ≥1 was found in 82% of the cases. ORR was 16%, median progression-free survival (PFS) was 2.1 months and OS was 8 months and such comparable to chemotherapy for this advanced situation [41].

Following these early data, Cohen et al. [14] conducted a multicentric phase III trial (Keynote-040) and randomized 495 R/M SCCHN patients progressing during or after platinum-based therapy to receive pembrolizumab or standard of care (SOC) chemotherapy consisting of either methotrexate, docetaxel or cetuximab. The group demonstrated that median OS was 8.4 months for the pembrolizumab cohort and 6.9 months for the SOC cohort (hazard ratio (HR) 0.80, 95% confidence interval (CI), 0.65–0.98, nominal *p* = 0.0161) and grade ≥3 AEs 13% vs. 36% in favor of the immunotherapy cohort. As such, the new substance proved to be both more effective and better tolerated than other monotherapy-regimens. Furthermore, in the tumor proportion score (TPS: Number of PD-L1 stained tumor cells/Total number of viable tumor cells) ×100) ≥50% subgroup (*n* = 129), PFS and OS were significantly prolonged through immunotherapy (PFS: HR 0.58, 95% CI, 0.39–0.86, *p* = 0.003; OS: HR 0.53, 95% CI, 0.35–0.81, *p* = 0.001). ORRs were 26.6% in the pembrolizumab group and 9.2% in the SOC group [42,43]. Despite initially missing the pre-specified primary endpoint (OS in the intention-to-treat cohort) the Keynote-040 trial led to the FDA approval (2016) of pembrolizumab for R/M SCCHN patients with a PD-L1 TPS of ≥50% after prior platinum-based therapy. 

In a next step, results of the second interim analysis of the Keynote-048 phase III randomized trial were presented at the annual meeting of the European Society for Medical Oncology (ESMO), 2018 in Munich, Germany [15]. A total of 825 patients with R/M SCCHN with ECOG PS 0–1 were randomized (1:1:1) to receive first line treatment consisting of either pembrolizumab alone, pembrolizumab + cis-/carboplatin + 5-fluorouracil (5-FU), or cis-/carboplatin + 5-FU and cetuximab (according to the EXTREME protocol [8]). The comparison of pembrolizumab alone versus EXTREME for the CPS ≥ 1 subgroup (*n* = 512) showed a higher median OS of 12.3 vs. 10.3 months (HR 0.78, 95% CI, 0.64–0.96, *p* = 0.0086), a lower ORR of 19.1% vs. 34.9% and a higher median duration of response (DOR) of 20.9 vs. 4.5 months, respectively for the pembrolizumab cohort. Treatment-related AEs grade 3–5 occurred in 16.7% and 69.0% in favor of the immunotherapy. For the CPS ≥ 20 subgroup (*n* = 255) median OS was prolonged to 14.9 vs. 10.7 months (HR 0.61, 95% CI, 0.45–0.83, *p* < 0.001). The comparison of pembrolizumab + chemotherapy vs. EXTREME showed a significant prolongation of OS for the combination of chemo- and immune therapy (13.0 vs. 10.7 months, HR 0.77, 95% CI, 0.63–0.93, *p* = 0.003). This trial is the first phase III comparison of immunotherapy and platinum-based chemotherapy and establishes pembrolizumab as first-line therapy for R/M SCCHN.

Recent studies assessing pembrolizumab for more specific indications include the ELDORANDO-trial, an ongoing phase II prospective randomized trial testing first line pembrolizumab 200 mg q3w vs. Methotrexate 40 mg/m^2^ body surface area (BSA) for elderly, frail or cisplatin ineligible patients with R/M SCCHN. Cisplatin ineligibility is defined as EGOG PS 2 and/or impaired renal function. The primary endpoint is one year-OS and recruitment is scheduled to be completed in 2021.

Nivolumab, another IgG4 PD-1 monoclonal antibody (BMS-936558, Bristol-Myers Squibb), received FDA approval in 2016 for R/M SCCHN with or without PD-L1 expression based on the results of the randomized, phase III CheckMate 141 trial by Ferris et al. [13]. In total, 361 patients were enrolled to receive either nivolumab 3 mg/kg q2w or SOC methotrexate, docetaxel or cetuximab (2:1 randomization). Median OS was significantly prolonged by nivolumab versus SOC (7.5 vs. 5.1 months, HR 0.70, 97.73% CI, 0.51–0.96, *p* = 0.01), whereas PFS was not affected. The response rate was 13.3% for nivolumab versus 5.8% for SOC and grade 3–4 toxicities occurred in 13.3% versus 35.1%, respectively. Toxicities included pneumonitis, dermatitis, and endocrine dysfunction. An updated long-term follow-up analysis indicated a 24-months OS rate of 16.9% for nivolumab and 6.0% for SOC, confirming the possibility for durable responses in a subgroup of patients, as it has been observed in malignant melanoma [44].

#### 2.2.2. Oligometastatic Disease

A phase II trial (NCT02684253) randomized patients with R/M SCCHN with or without prior platinum-based therapy and at least two distinct metastatic lesions between nivolumab 3 mg/kg q2w alone and nivolumab with stereotactic body radiotherapy applied to one lesion (SBRT, 9 Gray ×3). Primary outcome measure was an increase in the ORR of the non-irradiated lesion (abscopal effect) from 15% to 45%. However, ORR did not differ significantly between the two arms (30.8% for nivolumab mono, 25.9% for nivolumab + SBRT) [45].

The ongoing IMPORTANCE phase II trial randomizes patients (ECOG PS 0–1) with at least two distinct R/M lesions of SCCHN between pembrolizumab mono and pembrolizumab plus RT to 1–3 lesions. In case of CPS ≥ 1, pembrolizumab usage is allowed without prior platinum-based therapy. The main objective is to test the effect of local RT on systemic response to pembrolizumab.

The trial will assess the effect of local radiotherapy in addition to pembrolizumab. Recruitment started in 2018 and is estimated to be completed within 24 months with a total number of 130 patients.

#### 2.2.3. Combined PD-1 and CTLA-4 Inhibition

In an effort to intensify immune response, the recruiting Checkmate 714 double-blind, randomized (2:1) phase II trial applies first/second line nivolumab (q2w) and ipilimumab (IgG1 CTLA-4 monoclonal antibody, BMS-734016, Bristol-Myers Squibb, q6w) vs. nivolumab (q2w) and placebo (q6w) to R/M SCCHN patients. Primary outcome measure is ORR according to RECIST v1.1 [46]. The CheckMate 651 trial examines the efficacy of combined therapy with nivolumab and ipilimumab for first line R/M SCCHN treatment [47]. Enrollment started in 2017 and completion is estimated to be reached in 2020. Patients are randomized 1:1 between immunotherapy and the EXTREME protocol. Following a sequential strategy, OPTIM (NCT03620123) is an ongoing phase II trial delivering nivolumab for R/M SCCHN after prior platinum-based therapy, where in case of progression patients are randomized (1:1) to either receive nivolumab and ipilimumab (1 mg/kg BW q6w) or docetaxel (75 mg/m^2^ BSA q3w). Recruitment started in 2018 and estimated primary completion is awaited to be in 2022.

#### 2.2.4. PD-L1 Inhibition

Durvalumab (IgG1 PD-L1 monoclonal antibody, MEDI4736, AstraZeneca) has been approved for the treatment of locally-advanced non-small cell lung carcinoma after CRT [48]. No approval for SCCHN does exist so far, but several clinical trials were completed or are still ongoing. An initial phase I/II trial (Study 1108) enrolled 62 patients with R/M SCCHN in a basket trial design of different solid tumor entities, applying durvalumab (10 mg/kg q2w) for 12 months. The 12-month OS was 42%, ORR 11% and AEs ≥grade 3 occurred in 8% of the patients, who had a median of 3 prior systemic treatments [49,50]. To further investigate durvalumab in R/M SCCHN after prior platinum-based therapy, two phase II trials, HAWK and CONDOR, were conducted [51,52]. The HAWK single-arm trial applied durvalumab to patients with PD-L1 positive tumors (TPS ≥ 25%). Among the 111 patients, the ORR was 16.2%, median PFS and OS 2.1 and 7.1 months, respectively. Adverse events of any grade occurred in 57.1%, and of grade ≥ 3 in 8% of the cases. On the other hand, CONDOR enrolled patients with low/negative PD-L1 tumor expression (TPS < 25%). They were randomized in a 2:1:1 design to either receive durvalumab (20 mg/kg BW q4w) and tremelimumab (IgG2 CTLA-4 monoclonal antibody, AstraZeneca, 1 mg/kg BW q4w) for 4 cycles followed by durvalumab (10 mg/kg BW q2w), or durvalumab (10 mg/kg BW q2w) monotherapy, or tremelimumab (10 mg/kg BW q4w for 7 doses then every 12 weeks for 2 doses) monotherapy. Median OS did not differ between the groups and reached 7.6 months for combined modality therapy. The ORRs were 7.8% for durvalumab + tremelimumab, 9.2% for durvalumab, and only 1.6% for tremelimumab alone, demonstrating for the first time that CTLA-4 inhibition might contribute to enhanced toxicity and not to an improvement of response in SCCHN.

Two randomized phase III trials utilizing durvalumab completed recruitment recently. The EAGLE trial randomized patients 1:1:1 to receive second line durvalumab (10 mg/kg IV for up to 12 months) vs. tremelimumab (1 mg/kg IV) plus durvalumab (20 mg/kg IV for up to 12 months) vs. SOC (cetuximab, taxane, methotrexate, or fluoropyrimidine) [53]. In a press release by AstraZeneca from December 2018 it was stated that neither the combination of durvalumab and tremelimumab nor durvalumab alone prolonged OS vs. SOC (primary trial endpoint) [54].

In a similar design for platinum-eligible patients, the KESTREL trial randomized (2:1:1) to either first line flat-doses of tremelimumab 75 mg q4w and durvalumab 1500 mg q4w vs. durvalumab 1500 mg q4w vs. the EXTREME regimen. Primary endpoint is the efficiacy of durvalumab + tremelimumab vs. SOC in terms of OS [55]. First results are being awaited in 2019 [54] (Table 1).

A comparable agent, atezolizumab (IgG1 PD-L1 monoclonal antibody, MPDL3280A, Hoffmann-La Roche) was first applied to humans in a basket trial and administered as single agent to patients with locally advanced or metastatic solid malignancies or hematologic malignancies (total *n* = 661). Thirty-two SCCHN patients with locally advanced or metastatic disease were included, seven patients (22%) had primary tumors and 17 (53%) had ≥ 2 prior lines of therapy. A total of four (13%) experienced grade ≥ 3 AEs, ORR reached 22% and median OS was 6 months [56]. Based on these results, the IMvoke010 phase III trial for the primary situation is now recruiting [57].

### 2.3. Immune-Checkpoint Inhibition in the Curative Setting

Several phase III clinical trials for the primary treatment of LA-SCCHN are ongoing, but none reported definite results so far (Table 2). In this clinical scenario ICI are mostly investigated as augmentation for the validated treatment strategies RT/CRT and surgery (e.g., 70 Gy + cisplatin + placebo vs. cisplatin + ICI [58]). Preliminary results from the GORTEC 2015-01 prospective, randomized phase II trial (*n* = 133), which compared pembrolizumab vs. cetuximab in combination with RT for LA SCCHN indicated that treatment completion rate did not differ between the two groups, but ≥ grade 3 mucositis and radiodermatitis occurred significantly more often when RT was combined with cetuximab [59]. After initial safety trials [60], the results of ongoing large phase III trials presented below will give further important insights regarding feasibility and benefits of combined RT/CRT + ICI.

Lee et al. inaugurated a trial for patients receiving standard full-dose RT and cisplatin, randomized to either receive the PD-L1 antibody avelumab (IgG1 PD-L1 monoclonal antibody, L01X-C, Pfizer, 10 mg/kg BW) or placebo according to the following regime: Day 1 of the lead-in phase; Days 8, 25, and 39 of the CRT Phase; and q2w for 12 months during the maintenance phase. Lead-in avelumab is supposed to induce early immune response before/during CRT. Estimated primary completion is awaited in 2021 [58]. Another phase III trial REACH divides patients prior to randomization into fit (cisplatin eligible, *n* = 420) and unfit patients (defined as cisplatin ineligible, *n* = 268). Standard arm therapy comprises intensity-modulated RT (IMRT, 69.96 Gy, 33 fractions, simultaneously integrated boost) and either cisplatin (100 mg/m^2^ BSA, q3w) for the fit or cetuximab (400 mg/m^2^ BSA loading dose followed by 250 mg/m^2^ BSA q1w) for the unfit participants. Systemic therapy in the experimental arms consists of cetuximab plus avelumab (10 mg/kg BW q2w). A successful 1st step safety analysis was recently presented at the ASCO (American Society of Clinical Oncology) meeting 2018 [61]. Another trial investigating nivolumab + cisplatin vs. cisplatin for platinum eligible patients and nivolumab vs. cetuximab for platinum ineligible patients with primary IMRT 70/2 Gy was recently closed due to slow accrual without any results being published (NCT03349710). In the Keynote-412 trial patients with ECOG PS 0-1 are randomly assigned to receive pembrolizumab 200 mg q2w plus cisplatin-based CRT or placebo plus cisplatin-based CRT [62]. For LA SCCHN patients ineligible for cisplatin, the NRG-HN004 randomizes between primary RT (70 Gy IMRT) with either cetuximab or durvalumab (1500 mg FD q4w). The trial is actually suspended due to scheduled interim monitoring and the planned end date was prolonged from 2022 to 2025. Finally, following a “maintenance” strategy, another phase III trial evaluates the benefit of atezolizumab vs. placebo after definitive treatment of LA SCCHN patients. Patients receive atezolizumab 1200 mg FD q3w for up to one year with the aim to prevent local and distant disease recurrence [57].

In the peri-operative setting, the French NIVOPOSTOP trial aims to evaluate the benefit of adding nivolumab to CRT for patients with LA-SCCHN and ECOG PS 0–1, who receive postoperative CRT for R1 resection or extracapsular extension (ECE) of regional lymph node metastases. Patients are scheduled for standard IMRT 66/2 Gy and are randomized to either cisplatin (100 mg/m^2^ BSA on days 1, 22, 43 of RT) vs. cisplatin plus nivolumab (360 mg FD 3 weeks before RT and on days 1, 22 and 43 of RT). The German IMSTAR-HN trial compares neoadjuvant single time nivolumab 3 mg/kg within two weeks before surgery followed by standard surgical tumor resection including neck dissection plus postoperative RT/CRT with cisplatin, followed by post-adjuvant immunotherapy by either nivolumab or nivolumab plus ipilimumab (1 mg/kg q6w). In a similar setting the Kenyote-689 trial evaluates two cycles of pembrolizumab in the neoadjuvant setting followed by standard RT for low risk patients or cisplatin-based CRT plus pembrolizumab for high risk patients in the experimental arm vs. standard surgery plus postoperative RT/CRT in the control arm. This study is currently recruiting and is expected to be completed in 2026. Furthermore, two phase III trials utilizing durvalumab are currently launching (ADHERE) or are temporarily closed to accrual (NRG-HN004) in the primary treatment setting: ADHERE will test SOC postoperative CRT (66/2 Gy with cisplatin) for high risk (R1/ECE, HPV-negative) LA SCCHN together with durvalumab 1500 mg FD q4w or placebo. Recruitment is scheduled to start in the first half of 2019.

## 3. Targeted Therapies for SCCHN

### 3.1. EGFR-Inhibition: Mode of Action

Epidermal growth factor receptor (EGFR) is a transmembrane member of the ErbB receptor tyrosine kinase family. Following binding of one of its ligands EGF, transforming growth factor (TGF)-alpha or amphiregulin, receptor kinase activation results in the activation of the Ras/Raf/mitogen-activated protein kinase (MAPK), phosphoinositide 3-kinase (PI3K)/AKT and Janus kinase (JAK)/signal transducer and activator of transcription 3 (STAT3) pathways. These pathways facilitate cell proliferation, hamper apoptosis, promote angiogenesis, and activate invasion and metastasis [63,64] (Figure 3).

### 3.2. EGFR-Inhibition for SCCHN

EGFR is overexpressed in up to 90% of SCCHN and its overexpression has been shown to correlate with impaired prognosis and radiation-resistance [64,65]. Furthermore, preclinical studies indicated that EGFR-inhibition enhances radiation sensitivity [66].

EGFR-blockade was the first targeted therapy which was tested for treating SCCHN, combined with either cisplatin or RT [67,68]. Prospective phase II [69,70,71] and randomized phase III trials [7,8,65] led to the approval of cetuximab, a chimeric anti-EGFR-antibody for treating SCCHN both in the primary, curative setting, combined with RT and in the R/M setting combined with chemotherapy. To date, various chemotherapy/cetuximab combinations [8,72] remain the standard of care for the R/M situation [73] and cetuximab was applied as monotherapy for chemotherapy ineligible patients or combined with re-irradiation for locoregionally limited recurrent disease [74,75]. This standard is likely to change following recent findings in the field of immunotherapy demonstrating improved outcome of patients receiving ICI, at least when compared with cetuximab monotherapy [14]. Yet, cetuximab concomitant to RT remains standard of care in the primary, curative setting [73,76] for cisplatin-ineligible patients.

Interestingly, other drugs targeting the EGF-receptor, either by the humanized antibody panitumumab or by small molecule tyrosine-kinase inhibition failed to show any clinical benefit. The CONCERT-1 and 2 trials, two parallel conducted prospective, randomized phase II trials, failed to show any benefit either by triple-combination of panitumumab and CRT nor for panitumumab concomitant to RT versus cisplatin-based-CRT [77,78]. In order to explain these negative results, additional mechanisms of the cetuximab-mediated cytotoxicity have been proposed: besides inhibition of oncogenic pathways, cetuximab exerts its therapeutic activity by means of induction of an antibody dependent cell-mediated cytotoxicity [79], whereas the fully humanized antibody panitumumab lacks this feature. Regarding the triple combination of EGFR-inhibitors, RT and chemotherapy, cetuximab, which induces a G1 arrest [80] improves efficacy of RT against rapidly repopulating tumor cells during fractionated irradiation. However, this phenomenon may hamper the efficiency of the combination of RT and chemotherapy. For example, cisplatin exerts its radiosensitizing potential in proliferating cells, i.e., in the S/G2/M cell-cycle phases.

Furthermore, various efforts to directly inhibit the kinase domain of EGFR by small molecule inhibitors erlotinib and gefitinib also failed to improve results. Intriguingly, erlotinib showed good tolerance and promising results in phase I–II studies, mostly in the recurrent and metastatic setting [81,82,83,84,85], but also concomitant to RT or re-irradiation with curative intent [86,87,88,89]. Nevertheless, some combinations, e.g. with induction chemotherapy [90], proved to be extremely toxic or not effective [91] and one of the few randomized trials testing erlotinib in the definitive, curative treatment, failed to show any benefit when compared to standard CRT [92]. Gefitinib was tested in large, phase III randomized trials either as monotherapy compared to methotrexate [93] or as doublet together with docetaxel versus docetaxel plus placebo [94], with both studies revealing negative results. These data have stemmed optimism emerging from early phase II trials showing encouraging survival and tumor control rates for R/M diseases [95]. Additionally, Gregoire et al. were the first to demonstrate negative results, later confirmed by Ang et al. for cetuximab [96], and Mesia et al. for panitumumab [77] through triple-combination of gefitinib, cisplatin and RT [97]. Finally, combination of RT and gefitinib with paclitaxel seems to be toxic and not efficient [98,99]. In accordance with that, a randomized prospective trial on lapatinib, a novel EGFR/ErbB2-inhibitor approved for treating Her-2-positive breast cancer, also failed to demonstrate any benefit as concurrence and as maintenance to postoperative CRT in the curative setting [100].

Afatinib, a next generation irreversible EGFR-tyrosine kinase inhibitor, could have a place in SCCHN-treatment, at least in the second-line setting for metastatic patients. The “LUX-Head & Neck 1” study was a randomized phase III trial, demonstrating significantly improved PFS after afatinib when compared to methotrexate for this patient collective, although the median PFS was only 2.6 vs. 1.7 months for experimental and standard arm respectively, which makes the clinical impact of this result questionable (HR 0.80, 95% CI, 0.65–0.98, *p* = 0.030) [101]. By contrast, this seems not to be the case for the curative setting, when afatinib was used adjuvant to CRT in the “LUX-Head & Neck 2” and “LUX-Head & Neck 4” trials, which both have been terminated prematurely, due to not achieving the primary endpoint of improving disease free survival [102].

Unfortunately, more than 10 years after the first landmark trials, cetuximab remains the only targeted agent approved for the treatment of SCCHN. The future of cetuximab in SCCHN-treatment has been challenged recently, as the first phase III randomized trials indicating its inferiority compared to cisplatin even for the prognostically favorable HPV-positive-oropharyngeal cancer population were recently published [103,104]. Other recent efforts to implement cetuximab in more complex regimens, e.g. as a triplet simultaneous with CRT [96], or concomitant to RT after induction therapy [105] also failed to show any benefit compared to standard CRT. Thus, considering the improved outcome of immunotherapy, also in the palliative setting [14], it is likely that EGFR-inhibition will have a different role in the management of this disease in the near future. Possible new applications for cetuximab could be a combination with immunotherapy, as it demonstrates strong immunomodulatory effects [79], but also as second line treatment after failure of checkpoint inhibitors. Prospective trials for both indications are currently on the way [59] or could demonstrate the first encouraging results [106].

### 3.3. Targeted Therapies beyond EGFR-Inhibition

As EGFR-inhibition offered only marginal improvement of patient prognosis, the exploration of targeting alternative pathways was intensified. To this end, the best investigated target-pathway includes downstream effectors of the phosphatidylinositol-3-kinase PI3K/Akt/mammalian target of rapamycin signaling cascade, or shortly mTOR-pathway [107]. In 2017, Soulieres et al. published the results of the prospective, phase II, BERIL-1 trial, which indicated a significant improvement in terms of PFS and promising OS for the combination of paclitaxel and buparlisib, a PI3K-inhibitor compared with paclitaxel and placebo in the second line treatment of R/M disease [108]. Several trials utilizing mTOR-inhibitors like everolimus and temsirolimus have been conducted or are currently recruiting patients. Yet, most of the results so far are discouraging. In 2015, Massarelli et al. failed to demonstrate any benefit for the combination of everolimus and erlotinib, and Bauman et al. for temsirolimus and erlotinib for patients with R/M SCCHN [91,109]. The same holds true in another phase II trial, published for everolimus monotherapy [110], as well as for a simultaneous EGFR- and PI3K/mTOR-inhibition [111]. However, according to preclinical studies, mTOR-inhibition might manifest its full potential in combined regimen as shown by an enhancement of the effects of cytostatics like taxanes both in vitro and in vivo [112]. Indeed, a trial investigating such a combination of temsirolimus with carboplatin and paclitaxel showed manageable toxicity and considerable efficacy [113]. 

A phase II study implementing a similar combination-approach, without patient selection, failed to improve response rate or survival [111]. Thus, a careful selection of patients to benefit from similar strategies, by implementing novel predictive markers, could be of additional use [114]. This search is ongoing and even positive trials like the TEMHEAD-study of the German SCCHN-study group, a study investigating the effect of temsirolimus in platin- and cetuximab refractory patients, could not identify any reliable molecular predictive factors [115].

In summary, mTOR-inhibition does not show a clinical activity but is mostly well tolerated [116,117], even in combination with RT [118]. A recent study could also prove the down-regulation of down-stream signaling associated with clinical response following rapamycin application in the clinical setting [119]. Nevertheless, response rates remain modest and the optimal clinical setting and combination regimen remain to be established. An overview of current trials introducing different mTOR-inhibitors for SCCHN-treatment is depicted in Table 3.

Another targeted agent approved for the treatment of various solid malignancies like ovarian cancer and cervical cancer [122,123] is bevacizumab, an antibody targeting the vascular endothelial growth factor (VEGF). Up to date, predominately phase I-II data for this approach in head and neck cancer were published. In 2013, Holsinger et al. reported, that of five patients treated with temsirolimus with palliative intent, only the three who received concomitant bevacizumab showed objective response. This can be attributed either exclusively to the VEGF inhibition or to a synergistic anti-angiogenetic mechanism, as mTOR-targeting also attenuates the hypoxia-inducible factor HIF-1a [124]. Similar results have been observed in 46 patients treated with bevacizumab and erlotinib [83].

It is still not clear, if bevacizumab monotherapy is effective in the management of advanced or metastasized SCCHN and some authors report on enhanced toxicity after combination regimens with this agent: in a phase I trial evaluating the combination of bevacizumab with cisplatin, erlotinib and radiation, severe toxicity resulted in a study withdrawal [90] and the agent also seemed to enhance toxicity both in a phase II study investigating a non-platinum-based combination [125] and the rates of osteoradionecrosis in another publication [126]. Also, trials investigating the implementation of bevacizumab-combination regimens for recurrent and poor-prognosis SCCHN showed considerable and often unexpected toxicity [127]. However, other studies demonstrated acceptable rates of side effects when bevacizumab was applied together with intensity-modulated radiotherapy and carefully dosed cisplatin or cetuximab for patients with stages III-IV disease [125,128,129,130]. Accordingly, the feasibility of such treatments remains unclear as only one large phase III trial was completed [131]. In this trial, the addition of bevacizumab to a standard platinum doublet improved response rate and progression free survival, but not overall survival. Furthermore, severe bleeding-events were more common in the experimental arm. As anti-VEGF targeting enhances radiation response in preclinical models [132] and prospective data from a nasopharyngeal-carcinoma therapy [133] are also promising, further research seems justified. Selected running trials with bevacizumab are summarized in Table 4.

Finally, treatment with other multikinase-inhibitors, that have a proven anti-angiogenetic component, like sorafenib and sunitinib did not provide any substantial response or prolongation of survival in various phase II-studies [82,134,135,136], which led to a stepwise abandonment of further investigation with this substance class. Yet, Adkins et al. recently demonstrated a safe application and promising response rates for the combination of cetuximab with the new generation-angiogenesis inhibitor pazopanib in a phase Ib trial [137].

## 4. Conclusions

In summary, after an initial euphoria following approval of cetuximab as the first targeted agent for SCCHN almost 15 years ago, following advances in the field were disappointing and rare, but this may change by the advent of immunotherapy. As we have learned from the examples of EGFR-inhibition, similar agents do not necessarily show comparable results and less is sometimes more, as not every combination could fulfill the expectations. For example, the triple therapy consisting of radiotherapy, chemotherapy and cetuximab was not better than standard CRT and the same holds true when both PD-1 and CTLA-4 were targeted compared to anti-PD-1 monotherapy. Although current research [142,143] indicates the superiority of combined modality, every step should be undertaken carefully in order to avoid ineffective or toxic approaches. In times of personalized medicine, we should abandon “one size fits all” concepts, but rather address selected populations, based on both clinical features like elderly, frail, oligometastatic and molecular prognosticators/accurate biomarkers. Alternative strategies, that have yet to be studied in depth, both in terms of immune system modulation (like NK cell inhibitors, alternative checkpoints, adoptive T-cell therapies, vaccine approaches, etc.) and new molecular targets (e.g., CKD: cyclin depended kinase, WEE1 kinase) should also be considered [144,145]. At the same time, “traditional” treatments with proven efficacy like radiotherapy and chemotherapy should be integrated, as they open new possibilities, enhance the therapeutic spectrum of novel agents and could easily be combined with innovative technologies like nanoparticular drug transport [146]. Finally, RT demonstrates immunosensitizing effects and checkpoint-inhibition impacts on radiation sensitivity [10], however the best dose and fractionating schedule for combined approaches remains elusive. Thus, the future is challenging as systemic treatment for head and neck cancer remains the cutting edge of modern oncological research.

## Figures and Tables

**Figure 1 cancers-11-00472-f001:**
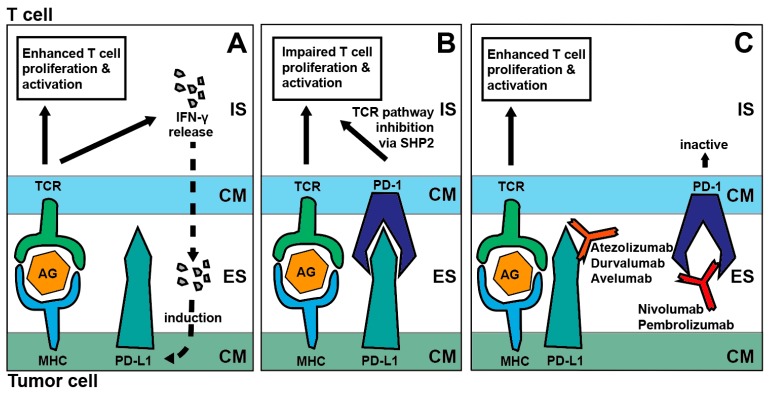
Mechanism of PD-1/PD-L1 blockade. (**A**): IFN-γ release upon TCR activation induces PD-L1 expression, (**B**): Interaction of PD-1/PD-L1 inhibits TCR signaling via SHP2, (**C**): Blockade of the PD-1/PD-L1 axis via atezolizumab, durvalumab, avelumab, nivolumab or pembrolizumab enhances T cell activation and proliferation; Abbreviations: IS—Intracellular space, CM—Cell membrane, ES—Extracellular space, TCR—T-cell receptor, MHC—Major histocompatibility complex I, AG—Antigen, IFN-γ—Interferon-γ, SHP2—Tyrosine phosphatase SHP-2, PD-1-Programmed cell death protein 1, PD-L1—Programmed cell death ligand 1.

**Figure 2 cancers-11-00472-f002:**
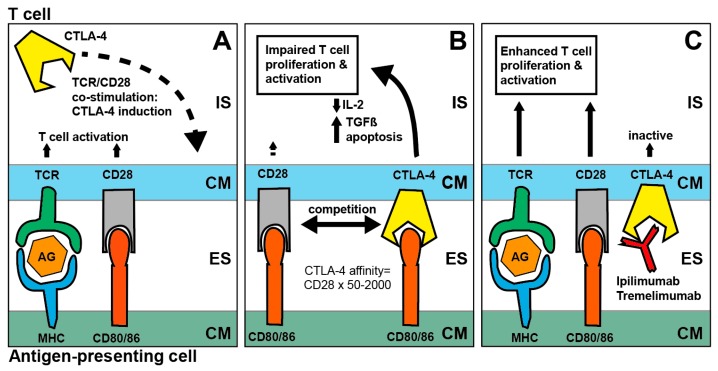
Mechanism of CTLA-4 blockade. (**A**): Co-stimulation of TCR/CD28 induces CTLA-4 receptor expression, (**B**): Interaction of CTLA-4 and CD80/86 inhibits IL-2 signaling, promotes apoptosis and the secretion of immunosuppressive i cytokines, such as TGFβ, (**C**): CTLA-4 receptor blockade by ipilimumab or tremelimumab enhances T cell activation and proliferation; Abbreviations: IS—Intracellular space, CM—Cell membrane, ES—Extracellular space, TCR—T-cell receptor, MHC—Major histocompatibility complex I, AG—Antigen, CTLA-4—Cytotoxic T-lymphocyte antigen 4, IL-2—Interleukin-2, TGFβ—Transforming Growth Factor beta.

**Figure 3 cancers-11-00472-f003:**
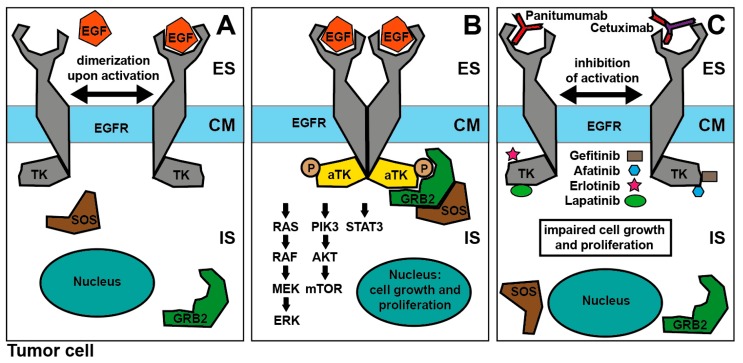
Mechanism of EGFR blockade. (**A**): Binding of EGF leads to receptor dimerization and activation of the intracellular tyrosine kinase activity and autophosphorylation, (**B**): Interaction of the intracellular tyrosine kinase with GRB2-SOS complex activates different pathways promoting cell growth and proliferation, (**C**): Extracellular EGFR receptor blockade by Cetuximab/Panitumumab and intracellular tyrosine kinase inhibitors gefitinib, afatinib, erlotinib and lapatinib impair cell growth and proliferation; Abbreviations: IS—Intracellular space, CM—Cell membrane, ES—Extracellular space, EGF—Epidermal growth factor, EGFR—Epidermal growth factor-receptor, TK—Tyrosine kinase, ATK—Activated tyrosine kinase, P—Phosphorylation, GRB2—Growth factor receptor-bound protein 2, SOS—Son of sevenless, RAS—G-protein Ras (Rat sarcoma), RAF—RAF-1 proto-oncogene serine/threonine-protein kinase, MEK—Mitogen-activated protein kinase kinase, ERK—Extracellular signal–regulated kinase, PIK3—Phosphoinositide 3-kinase, AKT—Serine/threonine-protein kinase B, mTOR—Mammalian target of Rapamycin kinase, STAT3—Signal transducer and activator of transcription 3.

**Table 1 cancers-11-00472-t001:** Selected trials including immune-checkpoint inhibition for recurrent/metastatic SCCHN.

PI/Author	Phase	Trial/NCT Number	Trial Design^#^	Substance	No. of Patients^‡^	Status
Colevas et al. [56]	Ia	**PCD4989g** NCT01375842	single-arm, multicentric	Atezolizumab*^,^^†^	32	complete
Seiwert et al. [36]	Ib	**Keynote-012** NCT01848834	single-arm, multicentric	Pembrolizumab^†^	60	complete
Segal et al. [49]	I/II	**Study 1108** NCT01693562	single-arm, multicentric	Durvalumab^†^	62	complete
Zandberg et al. [52]	II	**HAWK** NCT02207530	single-arm, multicentric	Durvalumab^†^	111	complete
Siu et al. [51]	II	**CONDOR** NCT02319044	randomized, multicentric	Durvalumab + Tremelimumab^†^ vs. Durvalumab vs. Tremelimumab	267	complete
Bauml et al. [41]	II	**Keynote-055** NCT02255097	single-arm, multicentric	Pembrolizumab^†^	171	complete
Grünwald et al.	II	**ELDORANDO** NCT03193931	randomized, multicentric	Pembrolizumab* vs. Methotrexate	e.e. 100	recruiting
Grünwald et al.	II	**OPTIM** NCT03620123	randomized, multicentric	Nivolumab + Ipilimumab^†^ vs.^¥^ Docetaxel	e.e. 280	recruiting
Fietkau et al.	II	**IMPORTANCE** NCT03386357	randomized, multicentric	Pembrolizumab^†^	e.e. 130	recruiting
McBride et al. [45]	II	NCT02684253	randomized, multicentric	Nivolumab + SBRT vs. Nivolumab	66	complete
Haddad et al. [46]	II	**Checkmate71**4 NCT02823574	double-blind, randomized, multicentric	Nivolumab + Ipilimumab vs, Nivolumab + Placebo	e.e. 315	recruiting
Cohen et al. [14]	III	**Keynote-040** NCT02252042	randomized, multicentric	Pembrolizumab^†^ vs. Methotrexate, Docetaxel or Cetuximab	495	complete
Burtness et al. [15]	III	**Keynote-048** NCT02358031	randomized, multicentric	Pembrolizumab* vs. Pembrolizumab + Cisplatin + 5FU vs. EXTREME	825	recruitment completed
Ferris et al. [13]	III	**CheckMate141** NCT02105636	randomized, multicentric	Nivolumab^†^ vs. Methotrexate, Docetaxel or Cetuximab	361	complete
Argiris et al. [47]	III	**CheckMate651** NCT02741570	randomized, multicentric	Nivolumab + Ipilimumab vs. EXTREME	e.e. 490	recruiting
Seiwert et al. [55]	III	**KESTREL** NCT02551159	randomized, multicentric	Durvalumab + Tremelimumab* vs. Durvalumab vs. EXTREME	823	recruitment completed
Ferris et al. [53]	III	**EAGLE** NCT02369874	randomized, multicentric	Durvalumab + Tremelimumab vs. Durvalumab vs. Cetuximab, Taxane, Methotrexate, or Fluoropyrimidine	720	recruitment completed

# All trials are open-labeled; * First line, † Second line or higher, ¥ Randomization after progression on Nivolumab mono after prior platinum-based therapy, ‡ Patients with squamous cell carcinoma of the head and neck—total number of patients in basket trials may exceed the given number; Abbreviations: PI—Principal investigator, e.e.—Estimated enrollment, 5FU—5-fluorouracil, EXTREME—Cis-/Carboplatin + 5-fluorouracil + Cetuximab, SBRT—Stereotactic body radiotherapy.

**Table 2 cancers-11-00472-t002:** Selected phase III clinical trials on immune-checkpoint inhibition for primary LA-SCCHN.

PI/Author	Trial/NCT Number	Substance and Treatment^#^	Primary Endpoint/Design	Estimated Enrollment^‡^/Primary Completion^¥^
Lee et al. [58]	Javelin Head and Neck 100 NCT02952586	70Gy RT + Cisplatin + Avelumab vs. Placebo	PFS/DB	N = 640 04/2021
GORTEC [61]	REACH NCT02999087	FIT: 70Gy RT + Cisplatin vs. Cetuximab + Avelumab UNFIT: 70Gy RT + Cetuximab vs. Cetuximab + Avelumab	PFS/OL	N = 640 10/2019
GORTEC	NIVOPOSTOP NCT03576417	66Gy PO RT Randomization: Cisplatin vs. Cisplatin + Nivolumab	DFS/OL	N = 484 12/2012
Busch et al.	IMSTAR-HN NCT03700905	Surgery +PO RT/CRT vs. Nivolumab + Surgery + PO RT/CRT + Nivolumab/Nivolumab + Ipilimumab	DFS/OL	N = 276 05/2024
Siu et al. [62]	Keynote-412 NCT03040999	70Gy RT + Cisplatin + Pembrolizumab vs. Placebo	EFS/DB	N = 780 04/2021
MSD	Keynote-689 NCT03765918	Pembrolizumab + Surgery + PO RT/CRT vs. Surgery + PO RT/CRT	mPR, EFS /OL	N = 600 01/2023
EORTC	ADHERE NCT03673735	66Gy PO RT + Cisplatin Randomization: Nivolumab vs. Placebo	DFS/DB	N = 650 07/2026
Mell et al.	NRG-HN004 NCT03258554	70Gy RT + Cetuximab vs. Durvalumab	DLT,PFS,OS/OL	N = 523 12/2025
Haddad et al. [57]	IMvoke010 NCT03452137	Atezolizumab vs. Placebo	EFS/DB	N = 400 08/2023

‡ Locally advanced squamous cell carcinoma of the head and neck, #All trials are open-labeled, randomized and multicentric, ^¥^ Last participant who received an intervention to collect final data for the primary outcome measure (dates according to clinicaltrials.gov accessed on 28 January 2019); Abbreviations: PI—Principal investigator, PFS—Progression-free survival, DFS—Disease-free survival, EFS—Event-free survival, OS—Overall survival, DLT—Dose-limiting toxicities, PO—Post-operative, RT—Radiotherapy, CRT—Chemoradiotherapy, GORTEC—Groupe Oncologie Radiotherapie Tete Et Cou, mPR—Major pathological response, DB—Double-blind, OL—Open-labeled, MSD—Merck Sharp & Dohme, EORTC—European Organization for Research and Treatment of Cancer.

**Table 3 cancers-11-00472-t003:** Overview of selected trials introducing mTOR inhibitors for head and neck cancer.

Author/ Trial/ NCT Number	Phase	No. of Patients	Setting	Regimen	Endpoint	Status
GERCOR NCT01333085	I/II	49	Curative-LA	Induction with Everolimus, Carboplatin, Paclitaxel followed by RT or surgery	I: MTD II: ORR	Completed No results
Saba et al. [116] NCT01283334	I/II	20	Palliative, 1st line	Carboplatin, Cetuximab, Everolimus	I: MTD II: PFS	MTD: 2.5 mg every other day, median PFS: 8.15 month
NCT01009346	I/II	9	Palliative, 1st line	Everolimus, Cetuximab and Cisplatin	I: MTD II: PFS	Terminated due to toxicity
Villaflor et al. [120] NCT01133678	I/II	94	Curative-LA	Induction Cisplatin, Paclitaxel, Cetuximab, Everolimus, followed by reduced-field RT	Tumor response rate	No benefit of everolimus
Massarelli et al. [91] NCT00942734	II	49	Palliative, 2nd line	Everolimus, Erlotinib	Tumor response rate	No benefit
NCT01016769	I/II	48	Palliative, 1st line	Temsirolimus, Paclitaxel, Carboplatin	I: dose finding II: ORR	Completed Results pending
TEMHEAD, Grünwald et al. [115] NCT01172769	II	40	Palliative, 1st/2nd line	Temsirolimus	PFS	Median PFS: 56 d Median OS: 152 d
Seiwert et al. [121] MAESTRO HN NCT01256385	II	86	Palliative 2nd line	Temsirolimus +/− Cetuximab (arms A vs. B)	PFS	Median PFS: 105 d in both arms 4-m-PFS: 41.3 vs 36.4%
Geiger et al. [110] NCT01051791	II	13	Palliative 1st line	Everolimus 10 mg/d	CBR	Terminated 28%; PFS: 1.5 mo, OS: 4.5 month
NCT01195922	II	37, only 16 treated	Curative-LA	Rapamycin (sirolimus) → surgery	%change in levels of pS6, pAKt473, Ki-67	Completed, no oncological results
NCT01015664	I/II	11	Palliative, 1st line	Cisplatin, Cetuximab, Temsirolimus	I: MTD II: PFS	Terminated

Abbreviations: LA—Locally advanced, RT—Radiotherapy, ORR—Objective response rate, MTD—Maximum tolerated dose, moMonths, d—Days, OS—Overall survival, PFS—Progression free survival, CBR—Clinical benefit rate (complete response, partial response, stable disease), GERCOR—Groupe Coopérateur Multidisciplinaire en Oncologie.

**Table 4 cancers-11-00472-t004:** Selected trials implementing VEGF-inhibition by bevacizumab.

Author/Trial/NCT Number	Phase	No. of Patients	Setting	Regimen	Endpoint	Status
Argiris et al. [138] **NCT00409565**	II	46	Palliative, 1 st/2 nd line	****Bevacizumab**** (15 mg/kg, q21) and Cetuximab	ORR	ORR: 16% Median PFS: 2.8 mo
**NCT00203905**	II	23/30	Curative-LA	CRT+FHX +/− **Bevacizumab**, randomized	PFS	Completed, no results
Fury et al. [129] **NCT00423930**	II	44	Curative-LA	IMRT + Cisplatin + **Bevacizumab**	2y-PFS	2-y-PFS: 75.9; 2-y-OS: 88%
Yao et al. [125] **NCT00281840**	I/II	30	Curative-LA	RT + weekly docetaxel 20 mg/qm+ **Bevacizumab** (5mg/kg biweekly)	Time to progression	3-y-PFS: 61.7%; 25/30 not completed
Argiris et al. [139] **NCT00703976**	II	80	Curative-LA	RT + Cetuximab + Pemetrexed +/− **Bevacizumab** (15 mg/kg, q21) randomized	2y-PFS	2y-PFS: 79% vs. 75%; 2y-OS: 91% vs. 87%
Argiris et al. [140] **NCT00222729**	II	42	Palliative, 1 st/2 nd line	Pemetrexed + **Bevacizumab** (15 mg/kg, q21)	Time to progression	Time to progression 5 mo; ORR: 30%
Cohen et al. **[83] NCT00055913**	I/II	58	Palliative, 1 st/2 nd line	Erlotinib + **Bevacizumab (q21)**	I: MTD; II: ORR	15 mg/kg, q21 ORR: 15.2%
Yoo et al. [126] **NCT00140556**	I	28	Curative-LA	Induction **Bevacizumab** + Erlotinib followed by CRT with Cisplatin, **Bevacizumab** + Erlotinib	Tumor resolution	25/26 patients
Hainsworth et al. [141] **NCT00392704**	II	60	Curative-LA	Induction Carboplatin, Paclitaxel, **Bevacizumab** (15 mg/kg, d 1+22) followed by CRT with Paclitaxel, Erlotinib, **Bevacizumab** (15 mg/kg, d 50+71)	2y-PFS	2y-PFS: 83%
ATHENA **NCT03818061**	II	Estimated 110	Palliative, 1st line	**Bevacizumab** (15 mg/kg, q21) + Atezolizumab	ORR	Not yet recruiting
**NCT00392665**	II	36/82	Palliative, 1 st/2 nd line	**Bevacizumab** + Erlotinib vs. Sulindac + Erlotinib randomized	PFS	9.38 vs 7.01 mo terminated due to slow accrual

Abbreviations: ORR—Overall response rate, PFS—Progression free survival, mo—Months, CRT—Chemoradiotherapy, FHX—5-fluorouracil + hydroxyurea, MTD—Maximum tolerated dose.

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
