# Peer review of "Targeted Therapies and Immune-Checkpoint Inhibition in Head and Neck Squamous Cell Carcinoma: Where Do We Stand Today and Where to Go?"

_cancers, 2019, doi:10.3390/cancers11040472_

Round 1
Reviewer 1 Report
This is an outstanding, comprehensive review of the rapid developments in immunotherapy for head and neck squamous cancer, and the current approaches to targeted therapy for this disease. The only major suggestion I have is that perhaps the review would benefit from a small section at the end of each of these topics summarizing the “where to go?” question. For example – for immunotherapy one could discuss a bit further the need for accurate biomarkers of response, and alternative strategies targeting the immune system that have yet to be studied in depth (eg. NK cell inhibitors, alternative checkpoints, adoptive T-cell therapies, vaccine approaches, etc.). For targeted therapy – any other new targets on the horizon? (Wee1 kinase inhibitors, others…).
However, I do realize this is a large undertaking, and if the manuscript in its existing form is satisfactory for the editors or there are page/word constraints limiting additional material, the manuscript is excellent as it stands.
Some minor suggestions:
Page 2 line 45 – the recurrence and metastasis are a bit high if referencing all diagnosed head and neck cancers, and the references cited focus on high-risk patients. Perhaps change the sentence to state that these rates are seen among patients with advanced/high risk resectable head and neck squamous cancer.
Page 5 line 196 – typographical error – should read “with or without”.
Author Response
Reviewer # 1:
Comment #1: The only major suggestion I have is that perhaps the review would benefit from a small section at the end of each of these topics summarizing the “where to go?” question. For example – for immunotherapy one could discuss a bit further the need for accurate biomarkers of response, and alternative strategies targeting the immune system that have yet to be studied in depth (eg. NK cell inhibitors, alternative checkpoints, adoptive T-cell therapies, vaccine approaches, etc.). For targeted therapy – any other new targets on the horizon? (Wee1 kinase inhibitors, others…).
Reply to the 1st comment: We would like to thank the reviewer for his/her valuable objection and fully agree that there also exist a variety of novel targets that have not yet been studied in depth. However, we consider that the detailed discussion of all these new approaches fare exceeds the intention of our manuscript, which already is about to reach the allowed word constraints. So, if the editors and the reviewers agree we would like to avoid adding new topics. However, we expanded the conclusion section by adding alternative strategies as suggested (lines 509-512).
Comment #2: Page 2 line 45 – the recurrence and metastasis are a bit high if referencing all diagnosed head and neck cancers, and the references cited focus on high-risk patients. Perhaps change the sentence to state that these rates are seen among patients with advanced/high risk resectable head and neck squamous cancer.
Reply to the 2nd comment: Again, we would like to thank you for this comment. We now have edited our text according to his/her suggestion (line 45).
Comment #3: Page 5 line 196 – typographical error – should read “with or without”.
Reply to the 3rd comment: The typographical error is corrected in the revised version of the manuscript (line 207).
Reviewer 2 Report
The authors of this review comprehensively describe the current development of checkpoint inhibitors and targeted therapies in SCCHN.
However, prior to publication some issues should be addressed:
· Line 170: The p value is incorrect and should be 0.0086. The OS was not similar but statistically significant different in the CPS>1 group as well, which should be stated.
· The important Keynote 689 phase III trial is missing (NCT03765918) and it is advised that the study goals are outlined in the article.
· Since the phase I trial of epacadostat plus pembro is mentioned (line 140), it should be stated that further development of this combination was stopped and the Keynote 669 phase III trial closed.
· Line 109-115: It might be worth mentioning that cytotoxic chemotherapy modulates the host immune system as well, since the Keynote 48 trial has already demonstrated that this concept seems to be successful.
· Line 390: It could be mentioned that the advantage of afatinib in the LUX-1 trial was just 2.6 months vs. 1.7 months (which is hardly a clinical meaningful difference)
· Line 404: The reviewer strongly disagrees with the statement that “EGFR-inhibition will play a minor or no role in the management of this disease in the near future”, because of the following reasons: 1) The authors stated correctly that cetuximab modulates the immune system as well and trials evaluating the effect of cetuximab plus immunotherapy are underway. The outcome of these trials are not predictable. 2) It is unclear, what will be the second line standard of care after immunotherapy failure in the R/M setting.There is preliminary evidence that response rates are improved with cetuximab plus chemo after checkpoint inhibitor failure (Saleh et al ASCO 2018). Therefore, revision of this statement is advised.
· Line 466 There are promising early and recent data of pazopanib plus cetuximab (Adkins et al. Lancet Oncology 2018) and the development of this multikinase inhibitor is pursued. This study could be added to table 4.
· Paragraph 3.3: PI3K pathway: The randomized phase II BERIL-1 study of buparlisib plus paclitaxel (Soulières et al Lancet Oncology 2017) should be cited. This study showed an OS of 10.4 months in the second line setting and demonstrated that PI3K/mTOR pathway inhibition might be a successful strategy. (Although the development of this compound was initially halted, the development seems to be continued now in SCCHN.)
Minor comment:
Several typos should be corrected such as line 181: Cisplatin not in capital letter or docetraxel in table 1 (OPTIM trial)
Author Response
Reviewer # 2:
Comment #1 Line 170: The p value is incorrect and should be 0.0086. The OS was not similar but statistically significant different in the CPS>1 group as well, which should be stated.
Reply to the 1st comment: We would like to thank the reviewer for the helpful comment and apologize for this mistake. We corrected the text according to his/her advice (line 179f).
Comment #2: The important Keynote 689 phase III trial is missing (NCT03765918) and it is advised that the study goals are outlined in the article.
Reply to the 2nd comment: We apologize for omitting this important trial. We now added this study both in chapter 2.3 (line 320ff) and in table 2.
Comment #3: Since the phase I trial of epacadostat plus pembro is mentioned (line 140), it should be stated that further development of this combination was stopped and the Keynote 669 phase III trial closed.
Reply to the 3rd comment: Although the Keynote-669 has never reached the planned accrual and no results are published yet, we agree with the reviewer that we should have mentioned it to complete the topic. Please find this addition in the revised manuscript (line 147ff).
Comment #4: Line 109-115: It might be worth mentioning that cytotoxic chemotherapy modulates the host immune system as well, since the Keynote 48 trial has already demonstrated that this concept seems to be successful.
Reply to the 4th comment: Although we already mentioned the results of the Keynote-48 trial in section 2.2.1, we agree that the biologic rationale behind this combination should also be edited here (line 114ff).
Comment #5: Line 390: It could be mentioned that the advantage of afatinib in the LUX-1 trial was just 2.6 months vs. 1.7 months (which is hardly a clinical meaningful difference)
Reply to the 5th comment: It is true, that the clinical advantage of this result is very questionable, so following the advice of the reviewer we added this consideration in the revised version of our manuscript (line 409ff).
Comment #6: Line 404: The reviewer strongly disagrees with the statement that “EGFR-inhibition will play a minor or no role in the management of this disease in the near future”, because of the following reasons: 1) The authors stated correctly that cetuximab modulates the immune system as well and trials evaluating the effect of cetuximab plus immunotherapy are underway. The outcome of these trials are not predictable. 2) It is unclear, what will be the second line standard of care after immunotherapy failure in the R/M setting.There is preliminary evidence that response rates are improved with cetuximab plus chemo after checkpoint inhibitor failure (Saleh et al ASCO 2018). Therefore, revision of this statement is advised.
Reply to the 6th comment:
We agree with the reviewer that this statement was somewhat exaggerated and accordingly, we now have changed this statement and also provided the two possibilities as mentioned by the reviewer as examples for a future impact of cetuximab (line 423ff).
Comment #7: Line 466 There are promising early and recent data of pazopanib plus cetuximab (Adkins et al. Lancet Oncology 2018) and the development of this multikinase inhibitor is pursued. This study could be added to table 4.
Reply to the 7th comment:
This is indeed a very interesting study that we failed to quote in the original manuscript. We now added a comment on this trial, not on table 4, as this table refers only to bevacizumab, but at the end of section 3.3. (line 491ff).
Comment #8: Paragraph 3.3: PI3K pathway: The randomized phase II BERIL-1 study of buparlisib plus paclitaxel (Soulières et al Lancet Oncology 2017) should be cited. This study showed an OS of 10.4 months in the second line setting and demonstrated that PI3K/mTOR pathway inhibition might be a successful strategy. (Although the development of this compound was initially halted, the development seems to be continued now in SCCHN.)
Reply to the 8th comment:
We would like to thank the reviewer for advising to this important study, utilizing a novel agent. We edited the text in section 3.3. and added this reference (line 432ff).
Comment #9: Several typos should be corrected such as line 181: Cisplatin not in capital letter or docetraxel in table 1 (OPTIM trial)
Reply to the 9th comment: In the revised version of the text, typos were corrected (line 192, table 1).
Round 2
Reviewer 2 Report
The authors addressed all issues satisfactorily.